# How unfair is private learning?

**Amartya Sanyal**[*1,3]  **Yaxi Hu**[*2]  **Fanny Yang**[3]

[1]ETH AI Center, ETH Zürich, Zürich, Switzerland.
[2]Department of Mathematics, ETH Zürich, Zürich, Switzerland.
[3]Department of Computer Science, ETH Zürich, Zürich, Switzerland.

## Abstract

As machine learning algorithms are deployed on sensitive data in critical decision making processes, it is becoming increasingly important that they are also private and fair. In this paper, we show that, when the data has a long-tailed structure, it is not possible to build accurate learning algorithms that are both private and results in higher accuracy on minority subpopulations. We further show that relaxing overall accuracy can lead to good fairness even with strict privacy requirements. To corroborate our theoretical results in practice, we provide an extensive set of experimental results using a variety of synthetic, vision (CIFAR-10 and CelebA), and tabular (Law School) datasets and learning algorithms.

## 1 INTRODUCTION

In recent years, reliability of machine learning algorithms have become ever more important due to their widespread use in daily life. Fairness and privacy are two instances of such reliability traits that are desirable but often absent in modern machine learning algorithms [10, 14, 39]As a result, there has been a flurry of recent works that aim to improve these properties in commonly used learning algorithms. However, most of these works discuss these two properties individually with relatively less attention paid to how they affect each other.

There is a multitude of definitions for privacy and fairness in their respective literatures. Perhaps the most widespread statistical notion of privacy is that of *Differential Privacy* [16] and its slightly relaxed variant, *Approximate Differential Privacy* [17]. Despite its marginally weaker privacy guarantees, *Approximate Differential Privacy* enjoys better theoretical guarantees in terms of statistical complexity for learning [7, 22]. It is also more widely used in practice [1, 49].

Thus, we always use approximate differential privacy in this paper and for the sake of brevity, refer to it as differential privacy (DP). Intuitively, DP limits the amount of influence any single data point has on the output of the DP algorithm. This ensures that DP algorithms do not leak information about whether any particular data point was given as input to the algorithm. While DP was initially popular as a theoretical construct, it has recently been put to practical use by large companies [20, 41] and governments [32] alike. Its popularity is largely due to its strong privacy guarantees, ease of implementation, and the quantitative nature of differential privacy.

There are many notions of fairness in machine learning [18, 26, 27]. Minority or worst group accuracy [15, 29] and its difference from the overall accuracy is a common notion of fairness used in recent works. We define this difference as *accuracy discrepancy* and use it to measure the degree of unfairness in this paper. However, we expect our results to translate to other related fairness metrics as well. Sagawa* et al. [37] observed that robust optimisation methods (under appropriate regularisations) obtain higher minority group accuracy but at the cost of a lower overall accuracy compared to vanilla training. Several other works have also observed this behavior in practice thereby suggesting a possible trade-off between overall accuracy and fairness metrics. Subsequent works including Goel et al. [25] and Menon et al. [34] have tried to minimise this trade-off.

While there have been a large body of works that aim to minimising the trade-off between accuracy and fairness [15, 25, 37] and between accuracy and privacy [17, 24], there is relatively few works that investigate the intersection of privacy, fairness, and accuracy. In this paper, we provide theoretical and experimental results to show that private and accurate algorithms are necessarily unfair. We further show that achieving privacy and fairness simultaneously leads to inaccurate algorithms.

**Contributions** Our main contributions can be stated as —

- In Theorem 1 and 2, we provide asymptotic lower

*Accepted for the 38ᵗʰ Conference on Uncertainty in Artificial Intelligence* (UAI 2022).

bounds for unfairness (accuracy discrepancy) of DP algorithms, that are accurate, showing that privacy and accuracy comes at the cost of fairness.

- In Theorem 3, we show that in a very strict privacy regime, fairness can be achieved at the cost of accuracy.

- In Section 3, we conduct experiments using multiple architectures on synthetic and real world datasets (CelebA, CIFAR-10, and Law School) to validate our theory.

**Related works**

It is now well understood that by imposing these additional conditions of DP more data are required to achieve high accuracy. A string of theoretical works [5, 9, 22] have shown that the sample complexity of learning certain concept classes privately and accurately can be arbitrarily larger than learning the same classes non-privately (i.e. with high accuracy but without privacy). On the other hand, it is easy to guarantee any arbitrary level of differential privacy if high accuracy is not desired. This can be achieved by simply composing the output of an accurate classifier with a properly calibrated *randomised response* mechanism [45]. This allows for a tradeoff between differential privacy and accuracy.

One of the most popular notions of fairness is *group fairness* that compares the performance of the algorithm on a minority group with other groups in the data. A popular instantiation of this, especially for deep learning algorithms, is comparing the accuracy on the minority group against the entire population [15, 25, 29, 37]. In the fairness literature, Buolamwini and Gebru [10], Raz et al. [36] shows extensively that this discrepancy is large between different groups of people for popular facial recognition systems.

DP-SGD [1] is widely used algorithm for implementing differentially private deep learning models. Bagdasaryan et al. [4] provides some experimental evidence that DP-SGD can have disparate impact on accuracy. Conversely, Chang and Shokri [11] shows, experimentally, that fairness aware machine learning algorithms suffer from less privacy. However, unlike these works, we provide theoretical results that are model agnostic and that discuss the dependance of the tradeoff on the subpopulation sizes and frequencies.

Cummings et al. [13] and Agarwal [2] were one of the first to consider the impact of privacy on fairness theoretically. They construct a distribution where any algorithm that is always fair and private will necessarily output a trivial constant classifier, thereby suggesting a tradeoff between fairness and privacy. However, there are multiple drawbacks with their work. First, their work only discusses pure differential privacy which is not only theoretically more restrictive than approximate differential privacy [6, 7, 22] but also rarely used in practice. Second, their proof heavily relies on it being *pure* differential privacy and the algorithm being *always* fair; and their proofs are not amenable to relaxations of

these assumptions. Further, they do not provide experiments to corroborate their theory perhaps due to the unrealistic requirements of pure DP. On the other hand, we look at approximate DP (which is a stronger result than pure DP), construct bounds for both fairness and error, and provide experimental results to support our theory. Perhaps, most closely related to our work is that of Feldman [21], who studies, mainly, the impact of memorisation on test accuracy for long-tailed distributions. However, neither does their work foray into differential privacy nor into fairness.

## 2 THEORETICAL RESULTS

The main contribution of our work is to provide a qualitative explanation for why and when differentially private algorithms cannot be simultaneously accurate and fair. Real world data distributions often contain a large number of subpopulations with very few examples in each of them and a few subpopulations with a large number of examples.

For example, Figure 1 (left) depicts the distribution of subpopulations in CelebA. Using the 40 attributes of the CelebA dataset [31], we partition the training set (of size $m = 160k$) into $2^{40}$ subpopulation bins. The blue shaded area shows the group of the subpopulations with large number of examples (probability mass greater than $\frac{1}{m}$) and the red shaded area corresponds to subpopulations which contain just one example in them. We refer to the subpopulations in the red area as *minority subpopulations* and the subpopulations in the blue area as *majority subpopulations*. This long-tailed structure over subpopulations have also been observed in Zhu et al. [51] in other vision datasets like the SUN [48] and PASCAL [19] datasets. Babbar and Schölkopf [3] observes this structure in extreme multilabel classification datasets like Amazon-670K [33] and Wikipedia-31k [8] datasets. Various other works [12, 30, 42, 44] have shown this structure in a range of datasets including eBird [40], Visual Genome [30], Pasadena trees [46], and iNaturalist [43].

### 2.1 PROBLEM SETUP

Mathematically, the large number of small subpopulations discussed above constitute the *long tail* of the distribution. We use this structure of data distributions to illustrate the tension between accuracy and fairness of private algorithms. For our theoretical results, we view each subpopulation as an element of a discrete set $X$ without any intrinsic structure such as distance. This distribution is inspired by the use of a similar distribution in Feldman [21].Next, we define a distribution over the subpopulations in $X$ that reflect the *long-tailed* structure.

**Definition 1** (($p, N, k$)-long-tailed distribution on $X$)**.** *Given $p \in (0, 1)$, $N \in \mathbb{N}$, and $1 < k \ll N$, define two groups (i) the group of majority subpopulations $X_1 \subset X$*

*Accepted for the 38$^{th}$ Conference on Uncertainty in Artificial Intelligence* (UAI 2022).

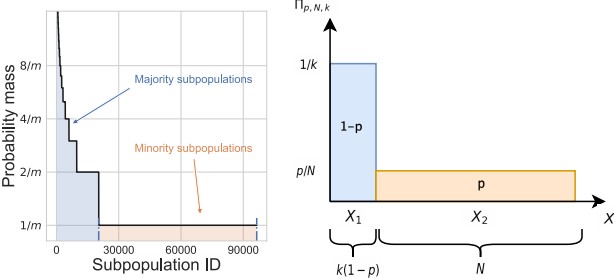

Figure 1: **(Left)** Illustration of the distribution of majority and minority subpopulations of CelebA. Here $m = 160k$ is the total size of the training set of CelebA. **(Right)** Illustration of $\Pi_{p,N,k}$.

*where $|X_1| = (1-p)k$ and (ii) the group of minority subpopulations $X_2 \subset X \setminus X_1$, where $|X_2| = N$. [1] Now, define the distribution $\Pi_{p,N,k}$ as*

$$\Pi_{p,N,k}(x) = \begin{cases} \frac{1}{k} & x \in X_1 \\ \frac{p}{N} & x \in X_2. \end{cases} \tag{1}$$

*We use the terms* group of majority subpopulations *and* majority group *interchangeably to denote $X_1$ and the terms* group of minority subpopulations *and* minority group *to refer to $X_2$ respectively.*

We provide an illustration of the distribution in Figure 1 (left). Here, $p$ denotes the total probability mass of the group of minority subpopulations under $\Pi_{p,N,k}$ and $N$ denotes the number of minority subpopulations. We let $N$ go to $\infty$ and treat $k$ as a constant. Thus, for the sake of simplicity, we remove $k$ from the notation of the distribution. Note that each minority subpopulation i.e. each element in $X_2$ has a probability mass of the order of $O\left(\frac{1}{N}\right)$ which is much smaller than $\frac{1}{k} = \Omega(1)$, i.e. the probability mass of each element in $X_1$.

Note that in the distribution for CelebA (Figure 1 (left)) the probability masses of the majority subpopulations not exactly equal to $\frac{1}{k}$ as in the distribution of Definition 1. However, all our results hold true even if different majority subpopulations have different probability masses as long as they satisfy $\Pi_{p,N}(x) = \Omega\left(\frac{1}{k}\right)$ for some $k = O(1)$ and all $x \in X_1$. We set them to $\frac{1}{k}$ only for simplicity of the theoretical results.

As we deal with a multiclass classification setup, we also define a label space $\mathcal{Y}$ and a function space $F$ of labelling functions. We also use $\mathcal{F}$ to represent a distribution on the function space $F$ and refer to this distribution as the label prior. Our results do not restrict the size of $\mathcal{Y}$ and hence, cover both binary and multi-class classification settings.

---

[1]WLOG we will assume that $k$ is such that $(1-p)k$ is an integer and if not, replace $k$ with the closest number such that $(1-p)k$ is an integer.

Finally, to generate a dataset of size $m$ from a $(p,N)$-long-tailed distribution $\Pi_{p,N}$ on $X$, first sample an unlabelled dataset $S = \{x_1, \cdots, x_m\}$ of size $m$ from $\Pi_{p,N}$. Then, generate the labelled dataset $S_f = \{(x_1, f(x_1)), \ldots, (x_m, f(x_m))\}$ using a labelling function $f \sim \mathcal{F}$. In all our theoretical results, we consider an asymptotic regime where $\frac{N}{m} \to c$ as $N, m \to \infty$. This is common in high-dimensional statistics where the number of dimensions often grows to $\infty$ along with the sample size. Intuitively, $c$ quantifies the hardness of the learning problem as it is invsersely proportional to the number of data points observed per minority subpopulation.

Next, we define the error and fairness measure of an algorithm on the distribution defined above. Consider a domain $X$, a label space $\mathcal{Y}$, a set of labelling functions $F$, a label prior $\mathcal{F}$, and a distribution $\Pi_{p,N}$ on $X$.

## 2.2 PRIVACY, ERROR, AND FAIRNESS

In the context of this paper, a differentially private (randomised) learning algorithm generates similar distributions over classifiers when trained on *neighbouring datasets*. Two datasets are neighbouring when they differ in one entry. Formally,

**Definition 2** (Approximate Differential Privacy [16, 17]). *Given any two neighbouring datasets $S, S'$, $\epsilon > 0$, and $\delta \in (0,1)$ an algorithm $\mathcal{A}$ is called $(\epsilon, \delta)$-differentially private if for all sets of outputs $\mathcal{Z}$, the following holds*

$$\mathbb{P}[\mathcal{A}(S) \in \mathcal{Z}] \le e^\epsilon \mathbb{P}[\mathcal{A}(S') \in \mathcal{Z}] + \delta.$$

Next, we define the error of an algorithm in our problem setup. For a randomised learning algorithm $\mathcal{A}$, a distribution $\Pi_{p,N}$, a label prior $\mathcal{F}$, we can define the error of the algorithm as follows

**Definition 3** (Error measure on $\Pi_{p,N}$). *The error of the algorithm $\mathcal{A}$ trained on a dataset of size $m$ from the distribution $\Pi_{p,N}$ with respect to a label prior $\mathcal{F}$ is*

$$\text{err}_m(\mathcal{A}, \Pi_{p,N}, \mathcal{F}) = \mathbb{E}[\mathbb{I}\{h(x) \ne f(x)\}] \tag{2}$$

*where $\mathbb{I}\{\cdot\}$ is the indicator function and the expectation is over $S \sim \Pi_{p,N}^m$, $f \sim \mathcal{F}$, $h \sim \mathcal{A}(S_f)$, and $x \sim \Pi_{p,N}$.*

Note that the error metric with an expectation over $\mathcal{F}$, was previously used in Feldman [21]. In fact, for the purpose of lower bounds on fairness, this is a stronger notion than the worst case $f \in F$ as, here, the lower bound is on the expectation which is stronger than a lower bound on the worst case. Next, we define the *accuracy discrepancy* of an algorithm, represented by $\Gamma$ over the distribution $\Pi_{p,N}$. For this purpose, for any $\Pi_{p,N}$, define the marginal distribution

*Accepted for the 38th Conference on Uncertainty in Artificial Intelligence* (UAI 2022).

on the group of *minority* subpopulations $X_2$ as

$$\Pi_{p,N}^2(x) = \begin{cases} \dfrac{\Pi_{p,N}(x)}{\sum_{x \in X_2} \Pi_{p,N}(x)} = \dfrac{\Pi_{p,N}(x)}{p} & x \in X_2 \\ 0 & x \notin X_2 \end{cases} \tag{3}$$

**Definition 4** (Accuracy discrepancy on $\Pi_{p,N}$). *For $X, \mathcal{F}, \Pi_{p,N}$, and $\Pi_{p,N}^2$ as defined above, the accuracy discrepancy of the algorithm $\mathcal{A}$ trained on a dataset of size $m$ on the distribution $\Pi_{p,N}$, with respect to the label prior $\mathcal{F}$, is*

$$\Gamma_m(\mathcal{A}, \Pi_{p,N}, \mathcal{F}) = \text{err}_m(\mathcal{A}, \Pi_{p,N}^2, \mathcal{F}) - \text{err}_m(\mathcal{A}, \Pi_{p,N}, \mathcal{F}). \tag{4}$$

*where $\text{err}_m(\cdot)$ is as defined in Definition 3.*

This notion of group fairness is similar to the notion of subgroup performance gap used in Goel et al. [25] and has also been implicitly used in multiple works Du et al. [15], Koh et al. [29], Sagawa* et al. [37] as discussed before. It has also been used in works related to the privacy [4, 11] and fairness literature [10, 36].

### 2.3 PRIVACY AND ACCURACY AT THE COST OF FAIRNESS

The theoretical results below use the definitions and notations described above and summarised in Table 1 in the supplementary. First, in Theorem 1, we show that there are distributions (within the family of distributions defined in Definition 1) where any accurate and approximately differentially private algorithm (with additional assumptions) is necessarily unfair. In Theorem 2, we relax some of the stronger assumptions and present a more general result. As discussed above, the dataset size $m$ and the number of minority subpopulations $N$ both simultaneously go to $\infty$ and the ratio $\frac{N}{m}$ is asymptotically equal to $\frac{N}{m} \sim c$. Throughout this section, we also use the notation $\text{err}(\mathcal{A}, \Pi_{p,N}, \mathcal{F}) = \lim_{m,N \to \infty} \text{err}(\mathcal{A}, \Pi_{p,N}, \mathcal{F})$ and $\Gamma(\mathcal{A}, \Pi_{p,N}, \mathcal{F}) = \lim_{m,N \to \infty} \Gamma(\mathcal{A}, \Pi_{p,N}, \mathcal{F})$ to denote the asymptotic limit for the error and the accuracy discrepancy metrics as $m, N \to \infty$.

**Theorem 1.** *For $\epsilon \in (0,1)$ and $\delta \in (0, 0.01)$, consider any $(\epsilon, \delta)$-DP algorithm $\mathcal{A}$ that does not make mistakes on subpopulations occurring more than once in the dataset. Then, there exists a family of label priors $\mathcal{F}$ where for any $\alpha \in (0,1)$, there exists $p \in (0, 1/2)$, $c > 0$ such that,*

$$\text{err}(\mathcal{A}, \Pi_{p,N}, \mathcal{F}) \leq \alpha \quad \text{and} \quad \Gamma(\mathcal{A}, \Pi_{p,N}, \mathcal{F}) \geq 0.5.$$

*where $\frac{N}{m} \to c$ as $N, m \to \infty$.*

A detailed version along with its full proof is presented in Appendix A.1. An immediate consequence of unfairness $\Gamma_m$ being greater than 0.5, coupled with the very small

error $\alpha$, is that the algorithm essentially behaves worse than random chance on the minority subpopulations thereby rendering the algorithm useless for these subpopulations.

**Proof sketch** Here we present a proof sketch and discuss the results. By Definition 1, the probability mass of each majority subpopulation is $\Omega(1)$ whereas the probability mass of each minority subpopulation is $O\left(\frac{1}{m}\right)$. Thus, for a large enough dataset (i.e. large $m$), we show that almost all majority subpopulations appear more than once and consequently, the algorithm in Theorem 1 makes very less mistakes on the majority subpopulations. As a result, the error and the accuracy discrepancy are both caused by mistakes, majorly, on minority subpopulations. We, then, count the number of subpopulations that do not appear or appear just once among the minority subpopulations and use that to provide the upper bounds for error and lower bound for unfairness (accuracy discrepancy). As these bounds are expressed in terms of $p$ and $c$, the proof then follows by showing the existence of $p, c$ that satisfy the inequalities in the theorem.

While Theorem 1 shows the existence of distributions under which private and accurate algorithms are necessarily unfair, in Theorem 2, we provide a quantitative lower bound for unfairness of private algorithms. In addition, we also generalise Theorem 1 to include a much broader set of algorithms. For this, we state two assumptions below. For any $\ell \in \mathbb{N}$, define $S^\ell$ to denote the set of examples that appear exactly $\ell$ times in $S$. Given $s_0 \in \mathbb{N}$ and $p_1, p_2 \in (0,1)$, we state that the algorithm $\mathcal{A}$ satisfies the assumptions A1 and A2 if the following conditions are satisfied by the $\mathcal{A}$ for all datasets $S$.

**Assumption on algorithm**

- (Accuracy) For all $\ell > s_0$ and $x \in S^\ell$,

$$\mathbb{P}_{f \sim \mathcal{F}, h \sim \mathcal{A}(S_f)}[h(x) \neq f(x)] \leq p_1 \tag{A1}$$

- (Privacy) For all $\ell \leq s_0$ and $x \in S^\ell$,

$$\mathbb{P}_{f \sim \mathcal{F}, h \sim \mathcal{A}(S_f)}[h(x) \neq f(x)] > 1 - p_2 \tag{A2}$$

Assumption A1 essentially requires the algorithm to have small overall train error. Note that since our domain is discrete, high training accuracy translates to high test accuracy in particular as the sample size approaches infinity. When $p_1$ is small, algorithm $\mathcal{A}$ obtains low training (and hence test) error on frequently occurring or *typical* data (i.e. $\ell \geq s_0$) On the other hand, Assumption A2 implies that the algorithm is incorrect on subpopulations that are rare in the training set, $\ell \leq s_0$, with a probability of at least $\geq 1 - p_2$. We refer to this assumption as the privacy assumption because for $(\epsilon, \delta)$-DP algorithms, it holds true for certain $s_0, p_2$ that depend on $\epsilon, \delta$, as discussed later in Lemma 1.

We note that the assumption in Theorem 1, that the algorithm does not make mistakes on subpopulations that appear more

*Accepted for the 38th Conference on Uncertainty in Artificial Intelligence* (UAI 2022).

than once is an instantiation of assumption A1 and A2 with the parameters $s_0 = 1$, $p_1 = 0$, and $p_2 = 1$. In the next result, we present a detailed result showing how unfairness of a DP algorithm varies with respect to any instantiations of the assumptions A1 and A2, privacy parameters, and distributional parameters. For easier interpretation, we show a simplified version in Theorem 2 and highlight the key takeaways, and provide a detailed version in Appendix A.2. Consider $X, \mathcal{F}$ as defined in Section 2.1.

**Theorem 2.** *For any $p \in (0, 1/2)$, $c > 0$ such that $p/c \leq 1$, consider the distribution $\Pi_{p,N}$ where $\frac{N}{m} \to c$ as $N, m$ goes to $\infty$. Also, for any $\epsilon, \delta > 0$, consider an $(\epsilon, \delta)$-DP algorithm $\mathcal{A}$ that satisfies assumptions A1 and A2 with $s_0 = o(m)$ and $p_1, p_2 \in (0, 1)$. Then, the accuracy discrepancy is lower bounded as follows*

$$\Gamma(\mathcal{A}, \Pi_{p,N}, \mathcal{F}) \geq (1-p)\gamma_0$$

*where $\gamma_0$ is some constant depending on $c, s_0, p, p_2, \epsilon, \delta$, and $\mathcal{F}$. Further, the error of the algorithm is upper-bounded as $\mathrm{err}(\mathcal{A}, \Pi_{p,N}, \mathcal{F}) \leq (1-p_1)p\alpha_0 + p_1$ and $\alpha_0$ depends on $c, s_0$ and $p$. If $s_0 \to \infty$ as $m \to \infty$, then in the asymptotic limit $c, m \to \infty$, $\gamma_0$ increases as $1 - O\left(\frac{e^{-cs_0}}{s_0\sqrt{c}}\right)$. As $p \to 0$, $\alpha_0$ increases as $1 - O\left(\sqrt{p}e^{-1/p}\right)$*

The detailed expressions of $c_0, \alpha_0$, and $\gamma_0$ (including its dependance on $\epsilon, \delta, p_1$, and $p_2$) can be found in Theorem 5 in Appendix A.2. We now briefly discuss how the theorem characterizes the effect of privacy and accuracy (via $s_0$) of the algorithm and the distribution and number of of subpopulations (via $p, c$ respectively) on the accuracy discrepancy. First note that when the ratio $c$ of the number of minority subpopulations with respect to the sample size is large, more minority subpopulations appear infrequently in the observed dataset and when $s_0$ is relatively large (but still $o(m)$) most infrequent subpopulations are misclassified. Indeed, theorem 2 indicates that as $c, s_0$ increase, $\gamma_0$ and hence unfairness increases, while the average error decreases. Intuitively, this is because minority subpopulations appear infrequently and the algorithm is less likely to memorise infrequent subpopulations. Therefore, the lower bound on unfairness increases with $c$ and $s_0$, and in the asymptotic limit of $m \to \infty$, accuracy discrepancy $\Gamma$ approaches $(1-p)$. [2]

**Remark 1.** *We note that one can omit Assumption A2 in Theorem 2 at the cost of restricting $s_0 < \frac{1}{2}\left(\min\left\{1, \frac{1}{\epsilon}\right\} \cdot \min\left\{\log\frac{1}{2\delta}, \log\frac{1}{2\|\mathcal{F}\|_\infty}\right\}\right)$ where $\|\mathcal{F}\|_\infty = \max_{x \in X, y \in \mathcal{Y}} \mathbb{P}_{f \sim \mathcal{F}}[f(x) = y]$. This follows from simple algebra on Equation (16) in the proof of Theorem 2 and Lemma 1 below. When the algorithm is private enough, i.e. $\epsilon, \delta$ are small enough, or when the label prior*

has a high entropy, $s_0$ can be large and hence leading to a larger lower bound on the accuracy discrepancy via Theorem 2.

Further, recall that $p$ quantifies the total probability mass of the group of minority subpopulations (see Figure 1). Hence, for a small $p$, error on minority subpopulations do not contribute significantly to the overall error despite causing a disproportionate increase to the marginal error of the minority group. As a result, as $p$ decreases, Theorem 3 states that the lower bound on unfairness increases as $1 - p - O\left(\sqrt{p}e^{-1/p}\right)$ while the upper bound for error decreases as $O\left(p - p^{3/2}e^{-1/p}\right) = O(p)$.

**Discussion of the assumptions** We now discuss how the privacy parameters $\epsilon, \delta$ of the DP algorithm lead to feasible parameters $s_0, p_2$ that appear in Assumptions A1 and A2 used in Theorem 2. We also provide an example of an algorithm that satisfies these assumptions. First, Lemma 1 shows that for all values of $\epsilon, \delta$, and $s_0$, there exists a value of $p_2$ that satisfies Assumption A2.

**Lemma 1.** *Let $\mathcal{F}$ be the uniform prior over all labelling functions and $S$ be any dataset. For any $(\epsilon, \delta)$-differentially private algorithm $\mathcal{A}$, for all $s_0 \in \mathbb{N}$, and for all subpopulations $x \in X$ that appear fewer than $s_0$ times in the dataset $S$, we have that*

$$\mathbb{P}_{f \sim \mathcal{F}, h \sim \mathcal{A}(S_f)}[h(x) \neq f(x)] > 1 - p_2$$

*for $p_2 = \frac{1}{2}\left(1 + \frac{1 + s_0 e^{-\epsilon}\delta}{1 + e^{-s_0 \epsilon}}\right)$.*

Please see Lemma 3 in the Appendix for a detailed version of Lemma 1. Lemma 3 also suggests how to extend this result (albeit with a different probability bound) to other label priors. Intuitively, the parameter $s_0$ in Lemma 1 (and assumption A2) represents the smallest frequency (in the observed data) of a subpopulation that is, with probability greater than $1 - p_2$, correctly classified by the algorithm $\mathcal{A}$. Lemma 1 proves that, for any choice of $\epsilon, p_2$ and sufficiently small $\delta$, there exists an $s_0$ such that any $(\epsilon, \delta)$-DP algorithm satisfies Assumption A2 with the parameters $s_0$ and $p_2$. In particular, for a fixed $p_2$, as $\epsilon$ decays, the probability lower bound is satisfied larger $s_0$ thereby showing that there is an inverse relationship between $s_0$ and $\epsilon$. We, thus, view $s_0$ as a "proxy" for the privacy of the algorithm in Theorem 2 and 3.

We now provide a differentially private algorithm that satisfies Assumptions A1 and A2. In particular, consider an algorithm $\mathcal{A}_\eta$ that accepts an $m$-sized dataset $S_f \in (X \times \mathcal{Y})^m$ and a noise rate $\eta \in \left(0, \frac{1}{2}\right)$ as input and outputs a dictionary matching every subpopulation in $X$ to a label of $\mathcal{Y}$. The algorithm first creates a dictionary where the set $X$ is the set of keys. In order to assign values to every key, it first randomly flips the label of every element in $S_f$ with probability $\eta$, then for every unique key in $S_f$,

---

[2]While the discussion here assumes the asymptotic limit for $c, s_0$, and $p$ as $m \to \infty$, our results in Appendix A.2 shows non-asymptotic dependence on these terms.

*Accepted for the 38th Conference on Uncertainty in Artificial Intelligence* (UAI 2022).

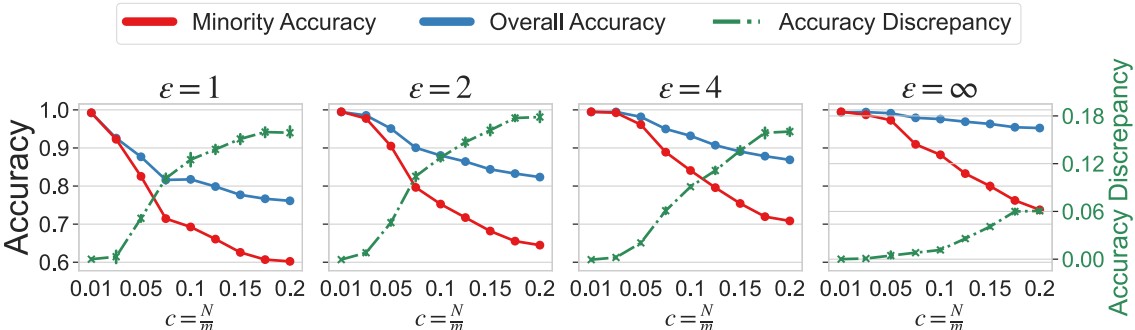

Figure 2: Each figure plots the accuracy discrepancy ($\Gamma$; higher is less fair) in green dashed line, the accuracy of the minority group with red, and the overall accuracy with blue on the y-axis and the parameter $c$ in the X-axis. The left most ($\epsilon = 1$) achieves the strictest level of privacy and the right most ($\epsilon = \infty$) is vanilla training without any privacy constraints. The two figures in between achieve intermediate levels of privacy. Here $p = 0.2$. Experiment for $p = 0.5$ is in Appendix B.1

the algorithm computes the majority label of that key in the flipped dataset and assigns that majority label to the corresponding key. For elements in $X$ not present in $S_f$, it assigns a random element from $\mathcal{Y}$. Lemma 2 provides privacy and accuracy guarantees for this algorithm.

**Lemma 2.** *The algorithm $A_\eta$ is $\left( O\left( \log\left( \frac{1}{\eta} \right) \right), 0 \right)$-DP as $\eta \to 0$. Further for any dataset $S_f$ and $s_0 \in \mathbb{N}$,*

- *if a subpopulation $x$ appears more than $s_0$ times in $S$,*
  $\mathbb{P}_{h \sim A_\eta(S_f)}[h(x) \neq f(x)] \leq e^{-s_0(1-2\eta)^2/8(1-\eta)}$ *and*

- *if a subpopulation $x$ appears less than $s_0$ times in $S$,*
  $\mathbb{P}_{h \sim A_\eta(S_f)}[h(x) \neq f(x)] \geq (4\eta(1-\eta))^{s_0/2} e^{-s_0}.$

*Equivalently, algorithm $A_\eta$ satisfies Assumption A1 with $p_1 = e^{-s_0(1-2\eta)^2/8(1-\eta)}$ and Assumption A2 with $p_2 = 1 - (4\eta(1-\eta))^{s_0/2} e^{-s_0}$.*

Lemma 1 and 2 are proved in Appendix A.2. Lemma 2 shows that for all $\epsilon > 0$, we can find an $\eta = O(e^{-\epsilon})$ such that $A_\eta$ is $(\epsilon, 0)$-differentially private. Further, this algorithm is more accurate on frequently occurring subpopulaitons and inaccurate on rare subpopulations, which aligns with Lemma 1. Hence, for any $\epsilon > 0$, there is an $\eta = O(e^{-\epsilon})$ such that the algorithm $A_\eta$ is $(\epsilon, 0)$-differentially private and is accurate on points appearing more than $\ell$ times with probability $1 - O\left(e^{-(1-2\eta)\ell}\right)$.

## 2.4 PRIVACY AND FAIRNESS AT THE COST OF ACCURACY

So far we have shown that under strict privacy and high average accuracy requirements on the algorithm, fairness

necessarily suffers. A natural question to ask is whether it is possible to sacrifice accuracy for fairness. As discussed in Section 2.3, increasing $s_0$ leads to higher error – in particular, we consider $s_0 = \Omega(m)$.

We present a simplified theorem statement here for easier interpretation and prove a more precise version in Appendix A.3 along with a discussion. In words, the theorem states that for very strict privacy parameters, fairness can be achieved at the cost of accuracy.

**Theorem 3.** *For any $p \in (0, 1/2)$, $c > 0$ such that $p/c \leq 1$, consider the distribution $\Pi_{p,N}$ where $N$ is the number of minority subpopulations. For any $\epsilon, \delta, \alpha > 0$, consider an $(\epsilon, \delta)$-DP algorithm $\mathcal{A}$ that satisfies assumptions (A1) and (A2) with $s_0 = \left( \frac{2-p}{2k(1-p)} \right) m + \alpha\sqrt{m}$ and some $p_2 \in (0, 1)$ where $\frac{N}{m} \to c$ as both $m, N \to \infty$. Then, ,*

$$\text{err}(\mathcal{A}, \Pi_{p,N}, \mathcal{F}) \geq c_1 p + (1 - p_2)(1 - p)\left( 1 - e^{-\frac{4(1-p)\alpha^2}{(2-p)^2}} \right)$$

$$\Gamma(\mathcal{A}, \Pi_{p,N}, \mathcal{F}) \leq (1 - p)(1 - p_2) e^{-\frac{4(1-p)\alpha^2}{(2-p)^2}} + p_2$$

*where $c_1$ is a constant depending on $\epsilon, \delta,$ and $\mathcal{F}$.*

The detailed expression of $c_1$ and the full proof can be found in Theorem 6 in Appendix A.3. We now briefly discuss how the theorem characterises the effect of privacy and accuracy (via $\alpha$) on fairness. When $\alpha$ is small, the algorithm is incorrect only on minority subpopulations. Thus, $\alpha$, essentially, characterises what fraction of majority subpopulations the algorithm is incorrect on. Theorem 3 shows that when $\alpha$ is large, the error increases and the unfairness decreases. Intuitively, this is because, with increasing $\alpha$, the algorithm is incorrect, not only on minority subpopulations, but also on *majority subpopulations* (due to Assumption A2). Thus, as

*Accepted for the 38$^{th}$ Conference on Uncertainty in Artificial Intelligence* (UAI 2022).

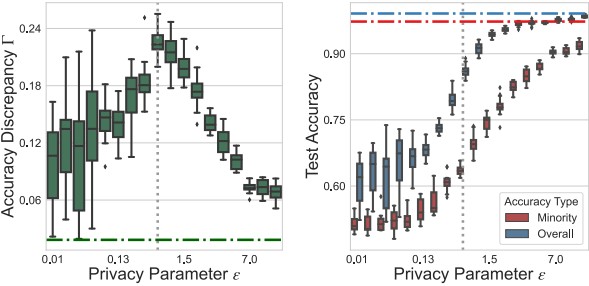

Figure 3: **Left:** Accuracy discrepancy (green) where the box plots reflect the variance when run several times. **Right:** Overall (blue boxes) and minority (red boxes) accuracies for varying $\epsilon$. The horizontal dashed line of different colors show the respective metrics for vanilla training without privacy constraints. The gray vertical dashed line marks the privacy parameter for which significant ($\geq 80\%$) overall test accuracy is achieved.

subpopulations of larger frequency gets misclassified due to increasing $\alpha$, the overall accuracy as well as the unfairness decreases.

## 3 EXPERIMENTAL RESULTS

In this section, we conduct experiments to support our theoretical results from Section 2. We note that our theoretical results are model-agnostic and to demonstrate the universality of our result, we conduct a broad set of experiments on both synthetic (in Section 3.1), and real world datasets (in Section 3.2 and 3.3), using multiple machine learning models including deep neural networks and random forests.

### 3.1 SYNTHETIC EXPERIMENTS

First, we look at a synthetic data distribution that closely emulates the data distribution we use in our theoretical results in Theorem 1 and 2. Given $N, k \in \mathbb{N}, c \in \mathbb{R}_{+}$, and $p \in (0, 0.5)$, we construct a continuous version of the long-tailed distribution $\Pi_{p,N,k}$ (Definition 1) on a domain $X$. First of all, since the domain $X$ is discrete, we can place each element on a vertex of a $O\left(\log\left(N\right)\right)$-dimensional hypercube. The continuous distribution we use in our experiments is a mixture of Gaussians where each Gaussian is centered around the vertices of the hypercube. In the experiments, we choose $k = 64, m = 10^4$, vary the ratio $c$ from $0.01$ to $0.2$, set the number of minority subpopulations to $N = mc$, and choose $p \in \{0.2, 0.5\}$. We train a five-layer fully connected neural network with ReLU activations using DP-SGD [1] for varying levels of $\epsilon$ while setting $\delta = 10^{-3}$. We refer to Appendix B for a more detailed description of the data distribution and the training algorithm.

**Unfairness aggravates with increasing number of minority subpopulations** As discussed in Section 2.1, increasing

the number of subpopulations compared to the number of samples via $c$ decreases accuracy on the minority subpopulations while the majority subpopulations remain unaffected. Figure 2 shows how increasing $c$ hurts fairness since the accuracy discrepancy (green dashed line) increases, most pronounced for small values $\epsilon$ (i.e. more private algorithms). This corroborates our theoretical results from Theorem 2 regarding the dependence of accuracy discrepancy on $c$. We further observe that the increase in unfairness is almost entirely due to the drop in the minority accuracy (red solid) whereas the overall accuracy (blue) stays relatively constant. This highlights our claim that, in the presence of strong privacy, fairness can be poor even when overall error is low.

**Privacy constraints hurt fairness for accurate models** In this section, we analyse the dependence of fairness on the privacy parameter $\epsilon$ for a fixed $c$. In Figure 3 (left), we plot the disparate accuracies $\Gamma$ for varying privacy parameter $\epsilon$ and Figure 3 (right) depicts the minority and overall accuracy as a function of $\epsilon$.

There are two distinct phases in the development of the accuracy discrepancy with increasing $\epsilon$ separated by the gray dashed line: For a very small $\epsilon$, the learned classifier is essentially a trivial classifier as evidenced by the very low overall accuracy ($\approx 60\%$). This is a trivial way of achieving fairness without learning an accurate classifier and is explained by Theorem 3 in our theoretical section. As the privacy restrictions are relaxed, the classifier becomes more accurate and less fair in the first phase.

The interesting regime is when classifier obtains decent overall accuracy ($\approx 80\%$) and is marked by the vertical gray dashed line. In the region to the right of the vertical dashed line, assumption $m = o\left(m\right)$ is fulfilled and Figure 3 (left) reflects the behavior as predicted in Theorem 2: *loosening privacy increases fairness or smaller $\epsilon$ implies larger accuracy discrepancy*.

### 3.2 EXPERIMENTS ON VISION DATASETS

In this section, we show that our claims resulting from Theorem 2 and 3 do not only hold in synthetic settings but can also be observed in real-world computer vision datasets. In particular, we conduct experiments on two popular computer vision datasets — CelebA and CIFAR-10. CelebA is a dataset of approximately $160k$ training images of dimension $178 \times 218$ and another $20k$ of the same dimension for testing. CIFAR-10 is a 10-class classification dataset where there are $50k$ training images and $10k$ test images of dimension $3 \times 32 \times 32$. For CIFAR-10, we use a ResNet-18 and for CelebA, we use a ResNet-50 architecture.

#### 3.2.1 Minority and majority subpopulations

In practice, datasets like CelebA and CIFAR-10 often do not come with a label of what constitutes a subpopulation.

*Accepted for the 38th Conference on Uncertainty in Artificial Intelligence* (UAI 2022).

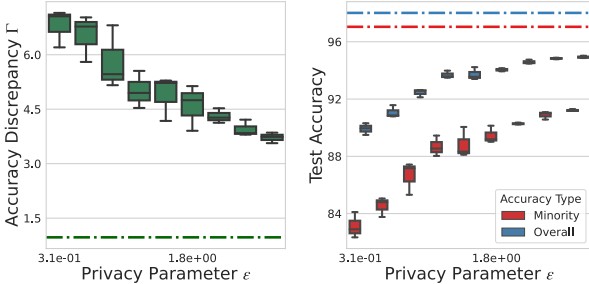 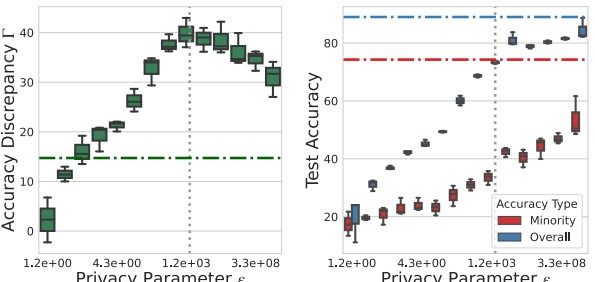

Figure 4: CelebA: **Left:** Accuracy discrepancy (green) where the box plots reflect the variance when run several times. **Right:** Overall (blue boxes) and minority (red boxes) accuracies for varying $\epsilon$. The horizontal dashed line of different colors show the respective metrics for vanilla training without privacy constraints.

Figure 5: CIFAR-10 **Left:** Accuracy discrepancy (green) where the box plots reflect the variance when run several times. **Right:** Overall (blue boxes) and minority (red boxes) accuracies for varying $\epsilon$. The horizontal dashed line of different colors show the respective metrics for vanilla training without privacy constraints. The vertical dashed line marks the $\epsilon$ for which significant ($\geq 75\%$) overall test accuracy is achieved.

In this section, we describe how we define the minority and majority subpopulations for CIFAR-10 and CelebA.

**CelebA** The CelebA dataset provides $40$ attributes for each image including characteristics like gender, hair color, facial hair etc. We create a binary classification problem by using the gender attribute as the target label. In addition, we use $11$ of the remaining $39$ binary attributes to create $2^{11}$ subpopulations and categorise each example into one of these $2^{11}$ subpopulations. Then, we create various groups of minority subpopulations by aggregating the samples of all the unique subpopulations that appear less than $s \in \{5, 10, 20, 40, 60, 80, 100\}$ times in the test set. The remaining examples constitute the majority group. In this section, we run experiments using $s = 40$. We report results for the other values of $s$ in Appendix B.2.

**CIFAR-10** Unlike the synthetic distribution and CelebA as described above, CIFAR-10 cannot be readily grouped into subpopulations using explicit attributes. However, recent works [38, 50] have shown the presence of subpopulations in CIFAR-10 in the context of influence functions and adversarial training respectively. We use the influence score estimates from Zhang and Feldman [50] to create the minority and majority subpopulations. Intuitively, we treat examples that are atypical i.e. unlike any other examples in the dataset as minority examples belonging to minority subpopulations; and examples that are *typical* i.e. similar to a significant number of other examples in the dataset as examples belonging to majority subpopulations.

To define these subpopulations, first, we sort the examples in the training set according to their self-influence [50]. We define all of those that surpass a threshold $\rho$ as minority populations. In order to find the samples belonging to each subpopulation $x$ in the test set, we search for images that are heavily influenced (influence score is greater than the threshold) by at least one of the samples in $x$ in the training set. In this section, we report results with $\rho = 0.1$. Other

values of $\rho$ show a similar trend and we plot results using $\rho = 0.01$ in Appendix B.3.

### 3.2.2 Privacy leads to worse fairness for accurate models

In this section, we use the above definitions of minority and majority groups to measure the impact of privacy on fairness (using Definition 4). Like Section 3.1, we measure both the accuracy discrepancy and the individual minority and overall accuracies.

**CelebA** Figure 4 plots the change in accuracy discrepancy, with respect to the $\epsilon$ parameter of differential privacy (smaller $\epsilon$ indicates stricter privacy). Figure 4 (left) shows that smaller $\epsilon$, with high accuracy (see Figure 4 (right)) implies a larger accuracy discrepancy. This aligns with our theoretical results from Section 2. Note that unlike Figure 3 (left), the accuracy discrepancy here monotonically decreases with increasing $\epsilon$ without exhibiting a two-phase behavior. Figure 4 (right) shows that, the reason why we do not observe the two-phase behavior is that throughout the range of observed $\epsilon$, we are in the regime of high accuracy.

**CIFAR-10** Figure 5 (left) plots the change of accuracy discrepancy $\Gamma$ with respect to the privacy parameter $\epsilon$. Interestingly, the results here exactly mimic those from the synthetic experiments in Figure 3, which are based on our theoretical setting. This indicates that our theoretical setting is indeed relevant for real world observations. Similar to the synthetic experiments, we observe two distinct phases in how the accuracy discrepancy changes with $\epsilon$.

For small values of $\epsilon$, Figure 5 (right) shows that the learned classifier is highly inaccurate. As discussed in Theorem 3, this is a trivial way to achieve fairness and this is reflected in Figure 5 (left). However, if we restrict ourselves to classi-

*Accepted for the 38th Conference on Uncertainty in Artificial Intelligence* (UAI 2022).

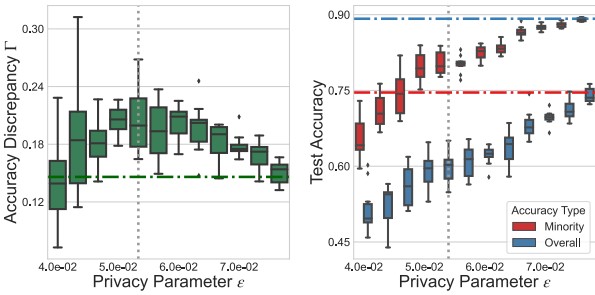

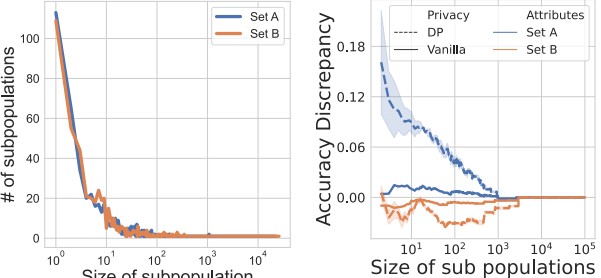

Figure 6: Law School—**Left:** Accuracy discrepancy (green) where the box plots reflect the variance when run several times. **Right:** Overall (blue boxes) and minority (red boxes) accuracies for varying $\epsilon$. The horizontal dashed line show the respective metrics for vanilla training without privacy constraints. The vertical dashed line marks the $\epsilon$ for which the test accuracy is largr ($\geq 80\%$).

Figure 7: **Left** Both sets of attributes induces a similar distribution over sizes of subpopulations on CelebA. **Right** Set A has high accuracy discrepancy for small sized subpopulations whereas Set B does not.

fiers with high average accuracy, marked by the area to the right of the vertical gray dashed line, Figure 5 (left) shows that accuracy discrepancy increases with decreasing $\epsilon$. This corresponds to the $s_0 = o(m)$ assumption in Theorem 2.

### 3.3 EXPERIMENTS ON TABULAR DATA

To show that our observations hold across a wider range of publicly used datasets, we next conduct similar experiments using tabular data. We run our experiments on the the Law school dataset [47] that has previously been used in fairness-awareness studies like Quy et al. [35]. It is a binary classification dataset with $21k$ data points and 12 dimensional features. Out of the 12 attributes, two binary attributes are used to obtain the minority group as defined in Quy et al. [35].

In contrast to previous experiments, we use random forest model from Fletcher and Islam [23] instead of neural networks as in Section 3.1 and 3.2. For our implementation, we use the publicly available code in Holohan et al. [28] with 10 trees and of a maximum depth 50. The results are plotted in Figure 6 and they show a similar two phase behavior as in our previous experiments with CIFAR-10.

Thus, all our experiments provide empirical evidence in support of the theoretical arguments in Section 2. The behavior is consistent across multiple kinds of datasets, machine learning models, and learning algorithms.

### 4 FUTURE WORK

The experimental results on CelebA in Section 3.2 shows that when the minority group is composed of small sized subpopulations, differential privacy requirements hurt the fairness of the algorithm. Here, we highlight that not all small-sized subpopulations are hurt equally in this process. Figure 7 shows that a different partition of CelebA

composed of similar sized populations do not show similar behaviours in terms of how accuracy discrepancy changes with sizes of subpopulations. We refer to the 11 attributes we chose to partition the testset for our experiments so far as *Set A* and, here, we choose another set of 11 attributes and refer to them as *Set B*. Figure 7 (left) shows that both Set A and Set B induces a very similar distribution over sizes of subpopulations on the test set. However, Figure 7 (right) shows that while the group of minority subpopulations induced by Set A suffers very high accuracy discrepancy from private training compared to vanilla training, Set B does not (see Appendix B.2 for more details on Set A and Set B). This indicates that, irrespective of sizes, private training hurts fairness disproportionately more for certain subpopulations compared to others. In particular, an interesting direction of further research is to investigate where these minority subpopulations that are worse-affected by private training intersects with the subpopulations that are relevant for the specific domain. While most past works [4, 11] have also used sizes of subpopulations to differentiate between disparately impacted subpopulations, this suggests that that is not always the case.

In this paper, we have shown theoretically that when the minority group in the data is composed of multiple subpopulations, a DP algorithm can achieve very low error but necessarily incurs worse fairness. Further, we corroborated our theoretical results with experimental evidence on synthetic and real world computer vision datasets. However, our model-agnostic results, that shed a rather pessimistic light on algorithmic fairness and differential privacy, only apply under certain distributional assumptions. It is possible that in some real-world datasets there are fair and private algorithms that achieve a more optimistic trade-off. This begs further research to develop fair and private algorithms that are closer to the pareto optimal frontier.

### Acknowledgements

AS is supported by the ETH AI Center and Hasler Stiftung.

*Accepted for the 38th Conference on Uncertainty in Artificial Intelligence* (UAI 2022).

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
