# OpenReview forum: "How unfair is private learning ?"
_auai.org/UAI/2022/Conference — UAI 2022 Oral_

### Official Review · Reviewer_bVkQ · 2022-03-16

**Q2(1) Originality/Novelty:** 3
**Q2(2) Significance/Impact:** 3
**Q2(3) Correctness/Technical Quality:** 3
**Q2(6) Clarity Of Writing:** 3
**Q6 Overall Score:** 7
**Q8 Confidence In Your Score:** 4

**Q1 Summary And Contributions:**

This is among the first few works that study the intersection of privacy, fairness, and accuracy. Based on the definitions of approximate differential privacy and minority group disparate accuracy, this work shows that when high accuracy is desired, fairness is sacrificed; and in a very strict privacy regime, fairness can be achieved at the cost of accuracy. The authors provide theoretical proofs and empirical results based on synthetic and real-world datasets to support their findings.

**Q2 Assessment Of The Paper:**

More detailed information regarding each of these aspects is given below:

**Q2(4) Quality Of Experiments (Optional):**

3: Good: The experimental evaluation is adequate, and the results convincingly support the main claims.

**Q2(5) Reproducibility:**

2: Fair: Key resources (e.g., proofs, code, data) are unavailable but key details (e.g., proof sketches, experimental setup) are sufficiently well-described for an expert to confidently reproduce the main results.

**Q3 Main Strengths:**

1. The authors look at the problems of privacy, fairness, and accuracy from a new perspective that can potentially lead to promising future research discussions in the community.
2. The revealed findings are interesting and provide new insights that may have some impact.
3. Findings are supported with strong theoretical proofs and empirical results.
4. Writings are clear and easy to follow.

**Q4 Main Weakness:**

I think to get more conclusive findings, the authors need to experiment with other popular fairness notions and consider other types of data, such as text. Some important findings in the paper are based on the results of one single dataset. Also, I am a bit lost trying to connect the "long-tailed data" with minority groups. Do they represent the same things in this work? Some explanations in the introduction can help. Do the authors plan to release their code and data?

**Q5 Detailed Comments To The Authors:**

See Q4.

**Q7 Justification For Your Score:**

I am familiar with the field and also carefully read the manuscript before I made my assessment.

**Q9 Complying With Reviewing Instructions:**

1: Yes.

---

### Official Review · Reviewer_HiAK · 2022-04-08

**Q2(1) Originality/Novelty:** 3
**Q2(2) Significance/Impact:** 2
**Q2(3) Correctness/Technical Quality:** 3
**Q2(6) Clarity Of Writing:** 2
**Q6 Overall Score:** 5
**Q8 Confidence In Your Score:** 2

**Q1 Summary And Contributions:**

This paper studies the relations between worst group accuracy, approximate differential privacy, and overall accuracy on long-tail data. With some assumptions on the learning algorithm, the authors theoretically show under a DP algorithm, how the joint error and the disparity in accuracy between groups would change when the distribution change in terms of the subpopulation. The findings are supported by sufficient experiments.

**Q2 Assessment Of The Paper:**

More detailed information regarding each of these aspects is given below:

**Q2(4) Quality Of Experiments (Optional):**

3: Good: The experimental evaluation is adequate, and the results convincingly support the main claims.

**Q2(5) Reproducibility:**

3: Good: Key resources (e.g., proofs, code, data) are available and key details (e.g., proofs, experimental setup) are sufficiently well-described for competent researchers to confidently reproduce the main results.

**Q3 Main Strengths:**

1. The research topic is interesting. Understanding the tradeoff between privacy, fairness, and accuracy is essential for the development of trustworthy algorithms.

2. The major conclusions from theoretical findings are clear.

3. Supportive experiments look sound and serve as a good explanation for the theorems. The figures fully show the trend along the three directions.

**Q4 Main Weakness:**

1. The presentation for this paper can be further improved. Some details are hard to follow. With many notations defined one or two pages ago, notation abuse, or the change of variables, and variables without intuitive explanation, it is not easy to understand a theoretical result in a short time. The organizations also can be reconsidered. For example, I don't quite follow Lemma 1 and 2 under the content 'Privacy at the cost of fairness.' Pointing out their relations to the current topics would be helpful.

**Q5 Detailed Comments To The Authors:**

Minor:
The authors are encouraged to simplify some notations. Probably a Table for notations is helpful. Align the notations with Fig. 1 (left) or more figures also help the reading.

**Q7 Justification For Your Score:**

The paper presents interesting findings on the relations between privacy, fairness, and overall accuracy under some clearly stated assumptions. My major concern is the presentation of theoretical findings and probably the authors can make them easier to follow.

**Q9 Complying With Reviewing Instructions:**

1: Yes.

---

### Official Review · Reviewer_iSn6 · 2022-04-12

**Q2(1) Originality/Novelty:** 2
**Q2(2) Significance/Impact:** 2
**Q2(3) Correctness/Technical Quality:** 3
**Q2(6) Clarity Of Writing:** 3
**Q6 Overall Score:** 5
**Q8 Confidence In Your Score:** 4

**Q1 Summary And Contributions:**

This work leverages the approximate differential privacy to investigate the interaction between two desirable properties of machine learning algorithms fairness and privacy. Theoretical results along with corresponding experimental evidence on synthetic and real datasets are provided.

**Q2 Assessment Of The Paper:**

More detailed information regarding each of these aspects is given below:

**Q2(4) Quality Of Experiments (Optional):**

2: Fair: The experimental evaluation is weak: important baselines are missing, or the results do not adequately support the main claims.

**Q2(5) Reproducibility:**

3: Good: Key resources (e.g., proofs, code, data) are available and key details (e.g., proofs, experimental setup) are sufficiently well-described for competent researchers to confidently reproduce the main results.

**Q3 Main Strengths:**

Important topic and interesting research angle.

Theoretical results corroborated by experimental results.

**Q4 Main Weakness:**

While admitting the importance of this research direction, claimed contributions have been unclear to me. Related to the stated contribution 1 and 2, the authors discussed the advantageous approximate differential privacy over differential privacy, how about advantages when understanding interactions between fairness and privacy based on approximate differential privacy and on differential privacy? Are these advantages mainly due to the use of approximate differential privacy or the authors' contributions if any?

The focus data is the long-tailed. Although synthetic data is long-tailed generated, I do not see the real world data shares such a property. In addition, any reason on exclusively using computer vision datasets? Many tabular datasets actually share the long-tailed characteristics which might resonate with the long-tailed focus of this paper. For concrete examples, check [1] A survey on datasets for fairness‐aware machine learning, DAMI.



**Q5 Detailed Comments To The Authors:**

Related work on state of the art regarding the trade-off between accuracy and fairness. In addition, non deep learning approaches such as random forests are widely used in practice, which is one motivation of using the employed framework, in comparison to deep learning methods, are ignored. Relevant discussion and comparison can strengthen the merit of this work.

**Q7 Justification For Your Score:**

Detailed in Q3-5.

**Q9 Complying With Reviewing Instructions:**

1: Yes.

---

### Official Review · Reviewer_6i7X · 2022-04-13

**Q2(1) Originality/Novelty:** 3
**Q2(2) Significance/Impact:** 3
**Q2(3) Correctness/Technical Quality:** 3
**Q2(6) Clarity Of Writing:** 3
**Q6 Overall Score:** 8
**Q8 Confidence In Your Score:** 2

**Q1 Summary And Contributions:**

The authors focus on fairness and privacy (Approximate Differential Privacy) (ADP), which are desirable instances of reliability in machine learning.
The contributions are the following :
- When the minority group in the data is composed of multiple subpopulations, ADP can achieve very low error but will necessarily result in poor fairness.
- The fairness decreases as the number of subpopulations increases.
- A strict privacy regime can entail good fairness but at the expense of accuracy.

**Q10 Ethical Concerns (Optional):**

No  potential ethical concerns raised.

**Q2 Assessment Of The Paper:**

More detailed information regarding each of these aspects is given below:

**Q2(4) Quality Of Experiments (Optional):**

4: Excellent: The experimental evaluation is comprehensive and the results are compelling.

**Q2(5) Reproducibility:**

4: Excellent: Key resources (e.g., proofs, code, data) are available and key details (e.g., proof sketches, experimental setup) are comprehensively described for competent researchers to confidently and easily reproduce the main results.

**Q3 Main Strengths:**

(i) The literature reports a large body of work that focuses on minimizing the trade-off between accuracy and fairness and between accuracy and privacy, there is relatively little work that focuses on the intersection between privacy, fairness, and accuracy.
(ii) For an easier interpretation, the authors care to show a simplified version of Theorem 1 (about the conflicting effects of accuracy and privacy on the one hand and fairness on the other hand) and Theorem 3 (questioning the sacrifice of accuracy for fairness), besides the detailed versions developed in Appendix. Similarly, a proof sketch of Theorem 2 is provided in the body of the text.
(iii) The experimental protocol developed on synthetic data and on CelebA and CIFAR10 visual datasets is convincing.

**Q4 Main Weakness:**

I do not see any.

**Q5 Detailed Comments To The Authors:**

Minor comments :
page 4, In the previous section, we considered assumptions that allow the algorithm to generate classifiers which make<<<s>>> mistakes on frequently occurring examples but with a low probability and make<<<s>>> mistakes on infrequent samples with a high probability.
Page 5, Theorem 2 and its proof <<<is>>> included in Appendix A.2.
Page 16 , I do not see the necessity of second line in (24).

**Q7 Justification For Your Score:**

see Q3
The authors of this theoretical paper do care to provide a didactical presentation.

**Q9 Complying With Reviewing Instructions:**

1: Yes.

---

### Decision · Program_Chairs · 2022-05-15

**Decision:**

Accept (Oral)

**Comment:**

Meta Review: The paper studies the intersection between fairness, privacy, and accuracy. Reviewers are overall positive about the novel insights that the paper provides. Minor concerns are well covered by the rebuttal.